# Dimension-Free Exponentiated Gradient

**Francesco Orabona**
Toyota Technological Institute at Chicago
Chicago, USA
`francesco@orabona.com`

## Abstract

I present a new online learning algorithm that extends the exponentiated gradient framework to infinite dimensional spaces. My analysis shows that the algorithm is implicitly able to estimate the $L_2$ norm of the unknown competitor, $U$, achieving a regret bound of the order of $\mathcal{O}(U \log(U\,T + 1))\sqrt{T})$, instead of the standard $\mathcal{O}((U^2 + 1)\sqrt{T})$, achievable without knowing $U$. For this analysis, I introduce novel tools for algorithms with time-varying regularizers, through the use of local smoothness. Through a lower bound, I also show that the algorithm is optimal up to $\sqrt{\log(UT)}$ term for linear and Lipschitz losses.

## 1 Introduction

Online learning provides a scalable and flexible approach for solving a wide range of prediction problems, including classification, regression, ranking, and portfolio management. These algorithms work in rounds, where at each round a new instance is given and the algorithm makes a prediction. After the true label of the instance is revealed, the learning algorithm updates its internal hypothesis. The aim of the classifier is to minimize the cumulative loss it suffers due to its prediction, such as the total number of mistakes.

Popular online algorithms for classification include the standard Perceptron and its many variants, such as kernel Perceptron [6], and $p$-norm Perceptron [7]. Other online algorithms, with properties different from those of the standard Perceptron, are based on multiplicative (rather than additive) updates, such as Winnow [10] for classification and Exponentiated Gradient (EG) [9] for regression.

Recently, Online Mirror Descent (OMD)[1] and has been proposed as a general meta-algorithm for online learning, parametrized by a regularizer [16]. By appropriately choosing the regularizer, most online learning algorithms are recovered as special cases of OMD. Moreover, performance guarantees can also be derived simply by instantiating the general OMD bounds to the specific regularizer being used. So, for all the first-order online learning algorithms, it is possible to prove regret bounds of the order of $\mathcal{O}(f(\boldsymbol{u})\sqrt{T})$, where $T$ is the number of rounds and $f(\boldsymbol{u})$ is the regularizer used in OMD, evaluated on the competitor vector $\boldsymbol{u}$. Hence, different choices of the regularizer will give rise to different algorithms and guarantees. For example, p-norm algorithms can be derived from the squared $L_p$-norm regularization, while EG can be derived from the entropic one. In particular for the Euclidean regularizer $\eta\sqrt{T}\|\boldsymbol{u}\|^2$, we have a regret bound of $\mathcal{O}(\sqrt{T}(\|\boldsymbol{u}\|^2/\eta + \eta))$. Knowing $\|\boldsymbol{u}\|$ it is possible to tune $\eta$ to have a $\mathcal{O}(\|\boldsymbol{u}\|\sqrt{T})$ bound, that is optimal [1]. On the other hand, EG has a regret bound of $\mathcal{O}(\sqrt{T \log d})$, where $d$ is the dimension of the space.

In this paper, I use OMD to extend EG to infinite dimensional spaces, through the use of a carefully designed time-varying regularizer. The algorithm, that I call Dimension-Free Exponentiated Gradient (DFEG), does not need direct access to single components of the vectors, rather it only requires

to access them through inner products. Hence, DFEG can be used with kernels too, extending for the first time EG to the kernel domain. I prove a regret bound of $\mathcal{O}(\|\boldsymbol{u}\| \log(\|\boldsymbol{u}\|T+1)\sqrt{T})$. Up to logarithmic terms, the bound of DFEG is equal to the optimal bound obtained through the knowledge of $\|\boldsymbol{u}\|$, but it does not require the tuning of any parameter.

I built upon ideas of [19], but I designed my new algorithm as an instantiation of OMD, rather than using an ad-hoc analysis. I believe that this route increases the comprehension of the inner working of the algorithm, its relation to other algorithms, and it makes easier to extend it in other directions as well. In order to analyze DFEG, I also introduce new and general techniques to cope with time-varying regularizers for OMD, using the *local smoothness* of the dual of the regularization function, that might be of independent interest. I also extend and improve the lower bound in [19], to match the upper bound of DFEG up to a $\sqrt{\log T}$ term, and to show an implicit trade-off on the regret versus different competitors.

## 1.1 Related works

Exponentiated gradient algorithms have been proposed by [9]. The algorithms have multiplicative updates and regret bounds that depend logarithmically on the dimension of the input space. In particular, they proposed a version of EG where the weights are not normalized, called EGU.

A closer algorithm to mine is the epoch-free in [19]. Indeed, DFEG is equivalent to theirs when used on one dimensional problems. However, the extension to infinite dimensional spaces is nontrivial and very different in nature from their extension to $d$-dimensional problems, that consists on running a copy of the algorithm independently on each coordinate. Their regret bound depends on the dimension of the space and can neither be used with infinite dimensional spaces nor with kernels.

Vovk proposed two algorithms for square loss, with regret bounds of $\mathcal{O}((\|\boldsymbol{u}\| + Y)\sqrt{T})$ and $\mathcal{O}(\|\boldsymbol{u}\|\sqrt{T})$ respectively, where $Y$ is an upper bound on the range of the target values [20]. A matching lower bound is also presented, proving the optimality of the second algorithm. However, the algorithms seem specific to the square loss and it is not possible to adapt them to other losses. Indeed, the lower bound I prove shows that for linear and Lipschitz losses a $\sqrt{\log(\|\boldsymbol{u}\|T)}$ term is unavoidable. Moreover, the second algorithm, being an instantiation of the Aggregating Algorithm [21], does not seem to have an efficient implementation.

My algorithm also shares similarities in spirit with the family of self-confident algorithms [2, 7, 15], in which the algorithm self-tunes its parameters based on internal estimates.

From the point of view of the proof technique, the primal-dual analysis of OMD is due to [15, 17]. Starting from the work of [8], it is now clear that OMD can be easily analyzed using only a few basic convex duality properties. See the recent survey [16] for a lucid description of these developments. The time-varying regularization for OMD has been explored in [4, 12, 15], but in none of these works does the negative terms in the bound due to the time-varying regularizer play a decisive role. The use of the local estimates of strong smoothness is new, as far as I know. A related way to have a local analysis is through the *local norms* [16], but my approach is better tailored to my needs.

## 2 Problem setting and definitions

In the online learning scenario the learning algorithms work in rounds [3]. Let $\mathbb{X}$ a Euclidean vector space[2], at each round $t$, an instance $\boldsymbol{x}_t \in \mathbb{X}$, is presented to the algorithm, which then predicts a label $\hat{y}_t \in \mathbb{R}$. Then, the correct label $y_t$ is revealed, and the algorithm pays a loss $\ell(\hat{y}_t, y_t)$, for having predicted $\hat{y}_t$, instead of $y_t$. The aim of the online learning algorithm is to minimize the cumulative sum of the losses, on any sequence of data/labels $\{(\boldsymbol{x}_t, y_t)\}_{t=1}^{T}$. Typical examples of loss functions are, for example, the absolute loss, $|\hat{y}_t - y_t|$, and the hinge loss, $\max(1 - \hat{y}_t y_t, 0)$. Note that the loss function can change over time, so in the following I will denote by $\ell_t : \mathbb{R} \to \mathbb{R}$ the generic loss function received by the algorithm at time $t$. In this paper I focus on linear prediction of the form $\hat{y}_t = \langle \boldsymbol{w}_t, \boldsymbol{x}_t \rangle$, where $\boldsymbol{w}_t \in \mathbb{X}$ represents the hypothesis of the online algorithm at time $t$.

**Algorithm 1** Dimension-Free Exponentiated Gradient.

**Parameters:** $0.882 \leq a \leq 1.109$, $L > 0$, $\delta > 0$.
**Initialize:** $\boldsymbol{\theta}_1 = \mathbf{0} \in \mathbb{X}$, $H_0 = \delta$
**for** $t = 1, 2, \ldots$ **do**
    Receive $\|\boldsymbol{x}_t\|$, where $\boldsymbol{x}_t \in \mathbb{X}$
    Set $H_t = H_{t-1} + L^2 \max\left(\|\boldsymbol{x}_t\|, \|\boldsymbol{x}_t\|^2\right)$
    Set $\alpha_t = a\sqrt{H_t}$, $\beta_t = H_t^{\frac{3}{2}}$
    **if** $\|\boldsymbol{\theta}_t\| == 0$ **then** choose $\boldsymbol{w}_t = 0$
    **else** choose $\boldsymbol{w}_t = \frac{\boldsymbol{\theta}_t}{\beta_t \|\boldsymbol{\theta}_t\|} \exp\left(\frac{\|\boldsymbol{\theta}_t\|}{\alpha_t}\right)$
    Suffer loss $\ell_t(\langle \boldsymbol{w}_t, \boldsymbol{x}_t \rangle)$
    Update $\boldsymbol{\theta}_{t+1} = \boldsymbol{\theta}_t - \partial \ell_t(\langle \boldsymbol{w}_t, \boldsymbol{x}_t \rangle)\boldsymbol{x}_t$
**end for**

We strive to design online learning algorithms for which it is possible to prove a relative *regret* bound. Such analysis bounds the *regret*, that is the difference between the cumulative loss of the algorithm, $\sum_{t=1}^{T} \ell_t(\langle \boldsymbol{w}_t, \boldsymbol{x}_t \rangle)$, and the one of an arbitrary and fixed competitor $\boldsymbol{u}$, $\sum_{t=1}^{T} \ell_t(\langle \boldsymbol{u}, \boldsymbol{x}_t \rangle)$. We will consider $L$-Lipschitz losses, that is $|\ell_t(y) - \ell_t(y')| \leq L|y - y'|$, $\forall y, y'$.

I now introduce some basic notions of convex analysis that are used in the paper. I refer to [14] for definitions and terminology. I consider functions $f : \mathbb{X} \to \mathbb{R}$ that are closed and convex. Given a closed and convex function $f$ with domain $S \subseteq \mathbb{X}$, its Fenchel conjugate $f^* : \mathbb{X} \to \mathbb{R}$ is defined as $f^*(\boldsymbol{u}) = \sup_{\boldsymbol{v} \in S}\left(\langle \boldsymbol{v}, \boldsymbol{u}\rangle - f(\boldsymbol{v})\right)$. The Fenchel-Young inequality states that $f(\boldsymbol{u}) + f^*(\boldsymbol{v}) \geq \langle \boldsymbol{u}, \boldsymbol{v}\rangle$ for all $\boldsymbol{v}, \boldsymbol{u}$. A vector $\boldsymbol{x}$ is a subgradient of a convex function $f$ at $\boldsymbol{v}$ if $f(\boldsymbol{u}) - f(\boldsymbol{v}) \geq \langle \boldsymbol{u} - \boldsymbol{v}, \boldsymbol{x}\rangle$ for any $\boldsymbol{u}$ in the domain of $f$. The differential set of $f$ at $\boldsymbol{v}$, denoted by $\partial f(\boldsymbol{v})$, is the set of all the subgradients of $f$ at $\boldsymbol{v}$. If $f$ is also differentiable at $\boldsymbol{v}$, then $\partial f(\boldsymbol{v})$ contains a single vector, denoted by $\nabla f(\boldsymbol{v})$, which is the gradient of $f$ at $\boldsymbol{v}$. Strong convexity and strong smoothness are key properties in the design of online learning algorithms, they are defined as follows. A function $f$ is $\gamma$-strongly convex with respect to a norm $\|\cdot\|$ if for any $\boldsymbol{u}, \boldsymbol{v}$ in its domain, and any $\boldsymbol{x} \in \partial f(\boldsymbol{u})$,

$$f(\boldsymbol{v}) \geq f(\boldsymbol{u}) + \langle \boldsymbol{v} - \boldsymbol{u}, \boldsymbol{x}\rangle + \frac{\gamma}{2}\|\boldsymbol{u} - \boldsymbol{v}\|^2 \ .$$

The Fenchel conjugate $f^*$ of a $\gamma$-strongly convex function $f$ is everywhere differentiable and $\frac{1}{\gamma}$-strongly smooth [8], this means that for all $\boldsymbol{u}, \boldsymbol{v} \in \mathbb{X}$,

$$f^*(\boldsymbol{v}) \leq f^*(\boldsymbol{u}) + \langle \boldsymbol{v} - \boldsymbol{u}, \nabla f^*(\boldsymbol{u})\rangle + \frac{1}{2\gamma}\|\boldsymbol{u} - \boldsymbol{v}\|_*^2 \ .$$

In the remainder of the paper all the norms considered will be the $L_2$ ones.

## 3 Dimension-Free Exponentiated Gradient

In this section I describe the DFEG algorithm. The pseudo-code is in Algorithm 1. It shares some similarities with the exponentiated gradient with unnormalized weights algorithm [9], to the self-tuning variant of exponentiated gradient in [15], and to the epoch-free algorithm in [19]. However, note that it does not access to single coordinates of $\boldsymbol{w}_t$ and $\boldsymbol{x}_t$, but only their inner products. Hence, we expect the algorithm not to depend on the dimension of $\mathbb{X}$, that can be even infinite. In other words, DFEG can be used with kernels as well, on contrary of all the mentioned algorithms above.

For the DFEG algorithm we have the following regret bound, that will be proved in Section 4.

**Theorem 1.** *Let* $0.882 \leq a \leq 1.109$, $\delta > 0$, *then, for any sequence of input vectors* $\{\boldsymbol{x}_t\}_{t=1}^T$, *any sequence of $L$-Lipschitz convex losses* $\{\ell_t(\cdot)\}_{t=1}^T$, *and any* $\boldsymbol{u} \in \mathbb{X}$, *the following bound on the regret holds for Algorithm 1*

$$\sum_{t=1}^{T} \ell_t(\langle \boldsymbol{w}_t, \boldsymbol{x}_t \rangle) - \sum_{t=1}^{T} \ell_t(\langle \boldsymbol{u}, \boldsymbol{x}_t \rangle) \leq \frac{4\exp(1 + \frac{1}{a})}{L\sqrt{\delta}} + a\|\boldsymbol{u}\|\sqrt{H_T}\left(\ln\left(H_T^{\frac{3}{2}}\|\boldsymbol{u}\|\right) - 1\right),$$

*where* $H_T = \delta + \sum_{t=1}^{T} L^2 \max(\|\boldsymbol{x}_t\|, \|\boldsymbol{x}_t\|^2)$.

The bound has a logarithmic part, typical of the family of exponentiated gradient algorithms, but instead of depending on the dimension, it depends on the norm of the competitor, $\|\boldsymbol{u}\|$. Hence, the regret bound of DFEG holds for infinite dimensional spaces as well, that is, it is dimension-free.

It is interesting to compare this bound with the usual bound for online learning using an $L_2$ regularizer. Using a time-varying regularizer $f_t(\boldsymbol{w}) = \frac{\sqrt{t}}{\eta}\|\boldsymbol{w}\|^2$ it is easy to see, e.g. [15], that the bound would be[3] $\mathcal{O}((\|\boldsymbol{u}\|^2/\eta + \eta)\sqrt{T})$. If an upper bound $U$ on $\|\boldsymbol{u}\|$ is known, we can use it to tune $\eta$ to obtain an upper bound of the order of $\mathcal{O}(U\sqrt{T})$. On the other hand, we obtain for DFEG a bound of $\mathcal{O}(\|\boldsymbol{u}\|\log(\|\boldsymbol{u}\|T+1)\sqrt{T})$, that is optimal bound, up to logarithmic terms, without knowing $U$. So my bound goes to constant if the norm of the competitor goes to zero. However, note that, for any fixed competitor, the gradient descent bound is asymptotically better.

The lower bound on the range of $a$ we get comes from technical details of the analysis. The parameter $a$ is directly linked to the leading constant of the regret bound; therefore, it is intuitive that the range of acceptable values must have a lower bound different from zero. This is also confirmed by the lower bound in Theorem 2 below.

Notice that the bound is data-dependent because it depends on the sequence of observed input vectors $\boldsymbol{x}_t$. A data-independent bound can be easily obtained from the upper bound on the norm of the input vectors. The use of the function $\max(\|\boldsymbol{x}_t\|, \|\boldsymbol{x}_t\|^2)$ is necessary to have such a data-dependent bound and it seems that it cannot be avoided in order to prove the regret bound.

It is natural to ask if the $\log$ term in the bound can be avoided. Extending Theorem 7 in [19], we can reply in the negative to this question. In particular, the following theorem shows that the regret of any online learning algorithm has a satisfy to a trade-off between the guarantees against the competitor with norm equal to zero and the ones against other competitors. A similar trade-off has been proven in the expert settings [5].

**Theorem 2.** *Fix a non-trivial vector space $\mathcal{X}$, a specific online learning algorithm, and let the sequence of losses be composed by linear losses. If the algorithm guarantees a zero regret against the competitor with zero $L_2$ norm, then there exists a sequence of $T$ vectors in $\mathcal{X}$, such that the regret against any other competitor is $\Omega(T)$. On the other hand, if the algorithm guarantees a regret at most of $\epsilon > 0$ against the competitor with zero $L_2$ norm, then, for any $0 < \eta < 1$, there exists a $T_0$ and a sequence of $T \geq T_0$ unitary norm vectors $\boldsymbol{z}_t \in \mathcal{X}$, and a vector $\boldsymbol{u} \in \mathcal{X}$ such that*

$$\sum_{t=1}^{T}\langle\boldsymbol{u}, \boldsymbol{z}_t\rangle - \sum_{t=1}^{T}\langle\boldsymbol{w}_t, \boldsymbol{z}_t\rangle \geq (1-\eta)\|\boldsymbol{u}\|\sqrt{\frac{1}{\log 2}}\sqrt{T\log\left(\frac{\eta\|\boldsymbol{u}\|\sqrt{T}}{3\epsilon}\right)} - 2 \,.$$

The proof can be found in the supplementary material. It is possible to show that the optimal $\eta$ is of the order of $\frac{1}{\log T}$, so that the leading constant approaches $\sqrt{\frac{1}{\log 2}} \approx 1.2011$ when $T$ goes to infinity. It is also interesting to note that an $L_2$ regularizer suffers a loss of $\mathcal{O}(\sqrt{T})$ against a competitor with zero norm, that cancels the $\sqrt{\log T}$ term.

# 4 Analysis

In this section I prove my main result. I will first briefly introduce the general OMD algorithm with time-varying regularizers on which my algorithm is based.

## 4.1 Online mirror descent and local smoothness

Algorithm 2 is a generic meta-algorithm for online learning. Most of the online learning algorithms can be derived from it, choosing the functions $f_t$ and the vectors $\boldsymbol{z}_t$. The following lemma, that is a generalization of Corollary 4 in [8], Corollary 3 in [4], and Lemma 1 in [12], is the main tool to prove the regret bound for the DFEG algorithm. The proof is in the supplementary material.

**Algorithm 2** Time-varying Online Mirror Descent
---
    **Parameters:** A sequence of convex functions $f_1, f_2, \ldots$ defined on $S \subseteq \mathbb{X}$.
    **Initialize:** $\boldsymbol{\theta}_1 = \mathbf{0} \in \mathbb{X}$
    **for** $t = 1, 2, \ldots$ **do**
        Choose $\boldsymbol{w}_t = \nabla f_t^*(\boldsymbol{\theta}_t)$
        Observe $\boldsymbol{z}_t \in \mathbb{X}$
        Update $\boldsymbol{\theta}_{t+1} = \boldsymbol{\theta}_t + \boldsymbol{z}_t$
    **end for**
---

**Lemma 1.** *Assume Algorithm 2 is run with functions $f_1, f_2, \ldots$ defined on a common domain $S \subseteq \mathbb{X}$. Then for any $\boldsymbol{w}_t', \boldsymbol{u} \in S$ we have*

$$\sum_{t=1}^{T} \langle \boldsymbol{z}_t, \boldsymbol{u} - \boldsymbol{w}_t' \rangle \le f_T(\boldsymbol{u}) + \sum_{t=1}^{T} \left( f_t^*(\boldsymbol{\theta}_{t+1}) - f_{t-1}^*(\boldsymbol{\theta}_t) - \langle \boldsymbol{w}_t', \boldsymbol{z}_t \rangle \right),$$

*where we set $f_0^*(\boldsymbol{w}_1') = 0$. Moreover, if $f_1^*, f_2^*, \ldots$ are twice differentiable, and $\max_{0 \le \tau \le 1} \|\nabla^2 f_t^*(\boldsymbol{\theta}_t + \tau \boldsymbol{z}_t)\| \le \lambda_t$, then we have*

$$f_t^*(\boldsymbol{\theta}_{t+1}) - f_{t-1}^*(\boldsymbol{\theta}_t) - \langle \boldsymbol{w}_t, \boldsymbol{z}_t \rangle \le f_t^*(\boldsymbol{\theta}_t) - f_{t-1}^*(\boldsymbol{\theta}_t) + \frac{\lambda_t}{2} \|\boldsymbol{z}_t\|^2 .$$

Note that the above Lemma is usually stated assuming the strong convexity of $f_t$, that is equivalent to the strong smoothness of $f_t^*$, that in turns for twice differentiable functions is equivalent to a global bound on the norm of the Hessian of $f_t^*$ (see Theorem 2.1.6 in [11]). Here I take a different route, assuming the functions $f_t^*$ to be twice differentiable, but using the weaker hypothesis of local boundedness of the Hessian of $f_t^*$. Hence, for twice differentiable conjugate functions, this bound is always tighter than the ones in [4, 8, 12]. Indeed, in our case, the global strong smoothness cannot be used to prove any meaningful regret bound.

We derive the Dimension-Free Exponentiated Gradient from the general OMD above. Set in Algorithm 2 $f_t(\boldsymbol{w}) = \alpha_t \|\boldsymbol{w}\| (\log(\beta_t \|\boldsymbol{w}\|) - 1)$, where $\alpha_t$ and $\beta_t$ are defined in Algorithm 1, and $\boldsymbol{z}_t = -\partial \ell_t(\langle \boldsymbol{w}_t, \boldsymbol{x}_t \rangle) \boldsymbol{x}_t$. The proof idea of my theorem is the following. First, assume that we are on a round where we have a local upper bound on the norm of the Hessian $f_t^*$. The usual approach in these kind of proof is to have a regularizer that is growing over time as $\sqrt{t}$, so that the terms $f_t^*(\boldsymbol{\theta}_t) - f_{t-1}^*(\boldsymbol{\theta}_t)$ are negative and can be safely discarded. At the same time the sum of the squared norms of the gradients will typically be of the order of $\mathcal{O}(\sqrt{T})$, giving us a $\mathcal{O}(\sqrt{T})$ regret bound (see for example the proofs in [4]). However, following this approach in DFEG we would have that the sum of norms of the squared gradients grows much faster than $\mathcal{O}(\sqrt{T})$. This is due to the fact that the global strong smoothness is too small. Hence I introduce a different proof method. In the following, I will show the surprising result that with my choice of the regularizers $f_t$, the terms $f_t^*(\boldsymbol{\theta}_t) - f_{t-1}^*(\boldsymbol{\theta}_t)$ and the squared norm of the gradient cancel out. Notice that already in [12, 13] it has been advocated not to discard those terms to obtain tighter bounds. Here the same terms play a major role in the proof and they are present thanks to the time-varying regularization. This is in agreement with Theorem 9 in [19] that rules out algorithms with a fixed regularizer to obtain regret bounds like Theorem 1.

It remains to bound the regret in the rounds where we do not have an upper bound on the norm of the Hessian. In these rounds I show that the norm of $\boldsymbol{w}_t$ (and $\boldsymbol{\theta}_t$) is small enough so that the regret is still bounded, thanks to the choice of $\beta_t$.

## 4.2 Proof of the main result

We start defining the new regularizer and show its properties in the following Lemma (proof in the supplementary material). Note the similarities with EGU, where the regularizer is $\sum_{i=1}^{d} w_i (\log(w_i) - 1)$, $\boldsymbol{w} \in \mathbb{R}^d, w_i \ge 0$ [9].

**Lemma 2.** *Define $f(\boldsymbol{w}) = \alpha \|\boldsymbol{w}\| (\ln(\beta \|\boldsymbol{w}\|) - 1)$, for $\alpha, \beta > 0$. The following properties hold*

- $f^*(\boldsymbol{\theta}) = \frac{\alpha}{\beta} \exp \frac{\|\boldsymbol{\theta}\|}{\alpha}$.

- $\nabla f^*(\boldsymbol{\theta}) = \frac{\boldsymbol{\theta}}{\beta\|\boldsymbol{\theta}\|}\exp\frac{\|\boldsymbol{\theta}\|}{\alpha}$.

- $\|\nabla^2 f^*(\boldsymbol{\theta})\|_2 \leq \frac{1}{\beta\min(\|\boldsymbol{\theta}\|,\alpha)}\exp(\frac{\|\boldsymbol{\theta}\|}{\alpha})$.

Equipped with a local upper bound on the Hessian of $f^*$, we can now use Lemma 1. We notice that Lemma 1 also guides us in the choice of the sequences $\alpha_t$. In fact if we want the regret to be $\tilde{\mathcal{O}}(\sqrt{T})$, $\alpha_t$ must be $\tilde{\mathcal{O}}(\sqrt{T})$ too.

In the proof of Theorem 1 we also use the following three technical lemmas, whose proofs are in the supplementary material. The first two are used to upper bound the exponential function with quadratic functions.

**Lemma 3.** *Let $M > 0$, then for any $\frac{\exp(M)}{M^2+1} \leq p \leq \exp(M)$, and $0 \leq x \leq M$, we have $\exp(x) \leq p + \frac{\exp(M)-p}{M^2}x^2$ .*

**Lemma 4.** *Let $M > 0$, then for any $0 \leq x \leq M$, we have $\exp(x) \leq 1 + x + \frac{\exp(M)-1-M}{M^2}x^2$.*

**Lemma 5.** *For any $p, q > 0$ we have that $\frac{2}{\sqrt{p}} - \frac{2}{\sqrt{p+q}} \geq \frac{q}{(p+q)^{\frac{3}{2}}}$.*

*Proof of Theorem 1.* In the following denote by $n(\boldsymbol{x}) := \max(\|\boldsymbol{x}\|, \|\boldsymbol{x}\|^2)$. We will use Lemma 1 to upper bound the regret of DFEG. Hence, using the notation in Algorithm 1, set $\boldsymbol{z}_t = -\partial\ell_t(\langle\boldsymbol{w}_t,\boldsymbol{x}_t\rangle)\boldsymbol{x}_t$, and $f_t(\boldsymbol{w}) = \alpha_t\|\boldsymbol{w}\|(\log(\beta_t\|\boldsymbol{w}\|) - 1)$. Observe that, by the hypothesis on $\ell_t$, we have $\|\boldsymbol{z}_t\| \leq L\|\boldsymbol{x}_t\|$. We first consider two cases, based on the norm of $\boldsymbol{\theta}_t$.

**Case 1:** $\|\boldsymbol{\theta}_t\| > \alpha_t + \|\boldsymbol{z}_t\|$.

With this assumption, and using the third property of Lemma 2, we have

$$\max_{0\leq\tau\leq1}\|\nabla^2 f_t^*(\boldsymbol{\theta}_t + \tau\boldsymbol{z}_t)\| \leq \max_{0\leq\tau\leq1}\frac{\exp\left(\frac{\|\boldsymbol{\theta}_t+\tau\boldsymbol{z}_t\|}{\alpha_t}\right)}{\beta_t\min(\|\boldsymbol{\theta}_t+\tau\boldsymbol{z}_t\|,\alpha_t)} \leq \frac{\exp\left(\frac{\|\boldsymbol{\theta}_t\|+\|\boldsymbol{z}_t\|}{\alpha_t}\right)}{\beta_t\alpha_t} .$$

We now use the second statement of Lemma 1. We have that $\frac{\lambda_t\|\boldsymbol{z}_t\|^2}{2} + f_t^*(\boldsymbol{\theta}_t) - f_{t-1}^*(\boldsymbol{\theta}_t)$ can be upper bounded by

$$\frac{\|\boldsymbol{z}_t\|^2}{2\alpha_t\beta_t}\exp\left(\frac{\|\boldsymbol{\theta}_t\|+\|\boldsymbol{z}_t\|}{\alpha_t}\right) + \frac{\alpha_t}{\beta_t}\exp\left(\frac{\|\boldsymbol{\theta}_t\|}{\alpha_t}\right) - \frac{\alpha_{t-1}}{\beta_{t-1}}\exp\left(\frac{\|\boldsymbol{\theta}_t\|}{\alpha_{t-1}}\right)$$

$$\leq \frac{\|\boldsymbol{z}_t\|^2}{2\alpha_t\beta_t}\exp\left(\frac{\|\boldsymbol{\theta}_t\|+\|\boldsymbol{z}_t\|}{\alpha_t}\right) + \frac{\alpha_t}{\beta_t}\exp\left(\frac{\|\boldsymbol{\theta}_t\|}{\alpha_t}\right) - \frac{\alpha_{t-1}}{\beta_{t-1}}\exp\left(\frac{\|\boldsymbol{\theta}_t\|}{\alpha_t}\right)$$

$$= \exp\left(\frac{\|\boldsymbol{\theta}_t\|}{\alpha_t}\right)\left(\frac{\|\boldsymbol{z}_t\|^2}{2aH_t^2}\exp\left(\frac{\|\boldsymbol{z}_t\|}{\alpha_t}\right) + \frac{a}{H_t} - \frac{a}{H_{t-1}}\right) . \tag{1}$$

We will now prove that the term in the parenthesis of (1) is negative. It can be rewritten as

$$\frac{\|\boldsymbol{z}_t\|^2}{2aH_t^2}\exp\left(\frac{\|\boldsymbol{z}_t\|}{\alpha_t}\right) + \frac{a}{H_t} - \frac{a}{H_{t-1}} = \frac{\|\boldsymbol{z}_t\|^2 H_{t-1}\exp\left(\frac{\|\boldsymbol{z}_t\|}{\alpha_t}\right) - 2a^2 H_{t-1}L^2 n(\boldsymbol{x}_t) - 2a^2 L^4(n(\boldsymbol{x}_t))^2}{2aH_t^2 H_{t-1}},$$

and from the expression of $\alpha_t$ we have that $\frac{\|\boldsymbol{z}_t\|}{\alpha_t} \leq \frac{1}{a}$, so we now use Lemma 3 with $p = 2a^2$ and $M = 1/a$. These are valid settings because $\frac{\exp(\frac{1}{a})}{\frac{1}{a^2}+1} \leq 2a^2 \leq \exp(\frac{1}{a})$, $\forall\, 0.825 \leq a \leq 1.109$, as it can be verified numerically.

$$\frac{\|\boldsymbol{z}_t\|^2}{2aH_t^2}\exp\left(\frac{\|\boldsymbol{z}_t\|}{\alpha_t}\right) + \frac{a}{H_t} - \frac{a}{H_{t-1}}$$

$$\leq \frac{\|\boldsymbol{z}_t\|^2 H_{t-1}\left(2a^2 + a^2(\exp(\frac{1}{a}) - 2a^2)\frac{\|\boldsymbol{z}_t\|^2}{\alpha_t^2}\right) - 2a^2 H_{t-1}L^2 n(\boldsymbol{x}_t) - 2a^2 L^4(n(\boldsymbol{x}_t))^2}{2aH_t^2 H_{t-1}}$$

$$\leq \frac{L^2\|\boldsymbol{x}_t\|^2 H_{t-1}\left(2a^2 + a^2(\exp(\frac{1}{a}) - 2a^2)\frac{L^2\|\boldsymbol{x}_t\|^2}{a^2 H_t}\right) - 2a^2 H_{t-1}L^2\|\boldsymbol{x}_t\|^2 - 2a^2 L^4\|\boldsymbol{x}_t\|^2}{2aH_t^2 H_{t-1}}$$

$$\leq \frac{L^4\|\boldsymbol{x}_t\|^4(\exp(\frac{1}{a}) - 4a^2)}{2aH_t^2 H_{t-1}} \leq 0, \tag{2}$$

where in last step we used the fact that $\exp(\frac{1}{a}) \leq 4a^2, \forall\, a \geq 0.882$, as again it can be verified numerically.

**Case 2:** $\|\boldsymbol{\theta}_t\| \leq \alpha_t + \|\boldsymbol{z}_t\|$.

We use the first statement of Lemma 1, setting $\boldsymbol{w}'_t = \boldsymbol{w}_t$ if $\|\boldsymbol{\theta}\| \neq 0$, and $\boldsymbol{w}'_t = \boldsymbol{0}$ otherwise. In this way, from the second property of Lemma 2, we have that $\|\boldsymbol{w}'_t\| \leq \frac{1}{\beta_t}\exp(\frac{\|\boldsymbol{\theta}_t\|}{\alpha_t})$. Note that any other choice of $\boldsymbol{w}'_t$ satisfying the the previous relation on the norm of $\boldsymbol{w}'_t$ would have worked as well.

$$
f_t^*(\boldsymbol{\theta}_{t+1}) - f_{t-1}^*(\boldsymbol{\theta}_t) = \frac{\alpha_t}{\beta_t}\exp\left(\frac{\|\boldsymbol{\theta}_{t+1}\|}{\alpha_t}\right) - \frac{\alpha_{t-1}}{\beta_{t-1}}\exp\left(\frac{\|\boldsymbol{\theta}_t\|}{\alpha_{t-1}}\right)
$$

$$
\leq \exp\left(\frac{\|\boldsymbol{\theta}_t\|}{\alpha_t}\right)\left(\frac{\alpha_t}{\beta_t}\exp\left(\frac{\|\boldsymbol{z}_t\|}{\alpha_t}\right) - \frac{\alpha_{t-1}}{\beta_{t-1}}\right) = a\exp\left(\frac{\|\boldsymbol{\theta}_t\|}{\alpha_t}\right)\frac{\exp\left(\frac{\|\boldsymbol{z}_t\|}{a\sqrt{H_t}}\right)H_{t-1} - H_t}{H_{t-1}H_t}. \quad (3)
$$

Remembering that $\frac{\|\boldsymbol{z}_t\|}{\alpha_t} \leq \frac{1}{a}$, and using Lemma 4 with $M = \frac{1}{a}$, we have

$$
H_{t-1}\exp\left(\frac{\|\boldsymbol{z}_t\|}{a\sqrt{H_t}}\right) - H_{t-1} - L^2 n(\boldsymbol{x}_t) \leq H_{t-1}\exp\frac{L\|\boldsymbol{x}_t\|}{a\sqrt{H_t}} - H_{t-1} - L^2\|\boldsymbol{x}_t\|^2
$$

$$
\leq H_{t-1}\left(1 + \frac{L\|\boldsymbol{x}_t\|}{a\sqrt{H_t}} + a^2\left(\exp\left(\frac{1}{a}\right) - 1 - \frac{1}{a}\right)\frac{L^2\|\boldsymbol{x}_t\|^2}{a^2 H_t}\right) - H_{t-1} - L^2\|\boldsymbol{x}_t\|^2
$$

$$
= \frac{LH_{t-1}\|\boldsymbol{x}_t\|}{a\sqrt{H_t}} + \left(\exp\left(\frac{1}{a}\right) - 1 - \frac{1}{a}\right)\frac{L^2 H_{t-1}\|\boldsymbol{x}_t\|^2}{H_t} - L^2\|\boldsymbol{x}_t\|^2
$$

$$
\leq \frac{LH_{t-1}\|\boldsymbol{x}_t\|}{a\sqrt{H_t}} + L^2\|\boldsymbol{x}_t\|^2\left(\exp\left(\frac{1}{a}\right) - 2 - \frac{1}{a}\right) \leq \frac{LH_{t-1}\|\boldsymbol{x}_t\|}{a\sqrt{H_t}}, \quad (4)
$$

where in the last step we used the fact that $\exp(\frac{1}{a}) - 2 - \frac{1}{a} \leq 0, \forall\, a \geq 0.873$, verified numerically. Putting together (3) and (4), we have

$$
f_t^*(\boldsymbol{\theta}_{t+1}) - f_{t-1}^*(\boldsymbol{\theta}_t) - \langle \boldsymbol{w}'_t, \boldsymbol{z}_t\rangle \leq \exp\left(\frac{\|\boldsymbol{\theta}_t\|}{\alpha_t}\right)\frac{L\|\boldsymbol{x}_t\|}{H_t^{\frac{3}{2}}} - \langle \boldsymbol{w}'_t, \boldsymbol{z}_t\rangle
$$

$$
\leq \exp\left(\frac{\|\boldsymbol{\theta}_t\|}{\alpha_t}\right)\frac{L\|\boldsymbol{x}_t\|}{H_t^{\frac{3}{2}}} + L\|\boldsymbol{w}'_t\|\|\boldsymbol{x}_t\| \leq \exp\left(\frac{\|\boldsymbol{\theta}_t\|}{\alpha_t}\right)\frac{L\|\boldsymbol{x}_t\|}{H_t^{\frac{3}{2}}} + \exp\left(\frac{\|\boldsymbol{\theta}_t\|}{\alpha_t}\right)\frac{L\|\boldsymbol{x}_t\|}{\beta_t}
$$

$$
= 2\exp\left(\frac{\|\boldsymbol{\theta}_t\|}{\alpha_t}\right)\frac{L\|\boldsymbol{x}_t\|}{H_t^{\frac{3}{2}}} \leq \frac{2\exp(1 + \frac{1}{a})L\|\boldsymbol{x}_t\|}{H_t^{\frac{3}{2}}}, \quad (5)
$$

where in the second inequality we used the Cauchy-Schwarz inequality and the Lipschitzness of $\ell_t$, in the third the bound on the norm of $\boldsymbol{w}'_t$, and in the last inequality the fact that $\|\boldsymbol{\theta}_t\| \leq \alpha_t + \|\boldsymbol{z}_t\|$ implies $\exp(\frac{\|\boldsymbol{\theta}_t\|}{\alpha_t}) \leq \exp(1 + \frac{1}{a})$. Putting together (2) and (5) and summing over $t$, we have

$$
\sum_{t=1}^{T}\left(f_t^*(\boldsymbol{\theta}_{t+1}) - f_{t-1}^*(\boldsymbol{\theta}_t) - \langle \boldsymbol{w}'_t, \boldsymbol{z}_t\rangle\right) \leq \sum_{t=1}^{T}\frac{2\exp(1 + \frac{1}{a})L\|\boldsymbol{x}_t\|}{H_t^{\frac{3}{2}}} \leq \frac{2}{L}\sum_{t=1}^{T}\frac{\exp(1 + \frac{1}{a})L^2\|\boldsymbol{x}_t\|}{(\sum_{j=1}^{t}L^2\|\boldsymbol{x}_t\| + \delta)^{\frac{3}{2}}}
$$

$$
\leq \frac{4\exp(1 + \frac{1}{a})}{L}\sum_{t=1}^{T}\left(\frac{1}{\sqrt{\sum_{j=1}^{t-1}L^2\|\boldsymbol{x}_t\| + \delta}} - \frac{1}{\sqrt{\sum_{j=1}^{t}L^2\|\boldsymbol{x}_t\| + \delta}}\right) \leq \frac{4\exp(1 + \frac{1}{a})}{L\sqrt{\delta}},
$$

where in the third inequality we used Lemma 5.

The stated bound can be obtained observing that $\ell_t(\langle \boldsymbol{w}_t, \boldsymbol{x}_t\rangle) - \ell_t(\langle \boldsymbol{u}, \boldsymbol{x}_t\rangle) \leq \langle \boldsymbol{u} - \boldsymbol{w}_t, \boldsymbol{z}_t\rangle$, from the convexity of $\ell_t$ and the definition of $\boldsymbol{z}_t$. $\qquad\square$

## 5 Experiments

A full empirical evaluation of DFEG is beyond the scope of this paper. Here I just want to show the empirical effect of some of its theoretical properties. In all the experiments I used the absolute loss,

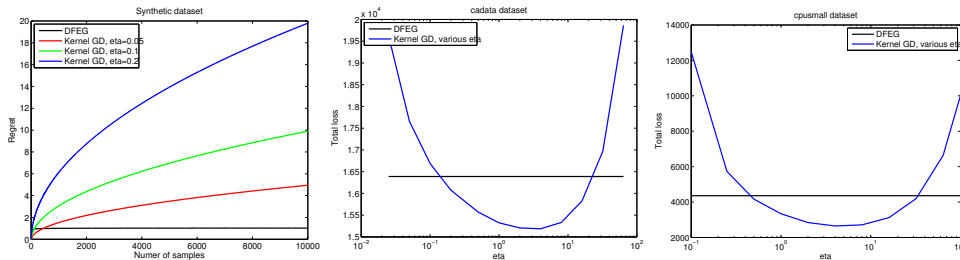

Figure 1: Left: regret versus number of input vectors on synthetic dataset. Center and Right: total loss for DFEG and Kernel GD on the *cadata* and *cpusmall* dataset respectively.

so $L = 1$, $a$ is set to the minimal value allowed by Theorem 1 and $\delta = 1$. I denote by Kernel GD the OMD with the regularizer $\frac{\sqrt{t}}{\eta}\|\boldsymbol{w}\|^2$.

First, I generated synthetic data as in the proof of Theorem 2, that is the input vectors are all the same and the $y_t$ is equal to 1 for the $t$ even and $-1$ for the others. In this case we know that the optimal predictor has norm equal to zero and we can exactly calculate the value of the regret. Figure 1(left) I have plotted the regret as a function of the number of input vectors. As predicted by the theory, DFEG has a constant regret, while Kernel GD has a regret of the form $\mathcal{O}(\eta\sqrt{T})$. Hence, it can have a constant regret only when $\eta$ is set to zero, and this can be done only with prior knowledge of $\|\boldsymbol{u}\|$, that is impossible in practical applications.

For the second experiment, I analyzed the behavior of DFEG on two real word regression datasets, *cadata* and *cpusmall*[4]. I used the Gaussian kernel with variance equal to the average distance between training input vectors. I have plotted in Figure 1(central) the final cumulative loss of DFEG and the ones of GD with varying values of $\eta$. We see that, while the performance of Kernel GD can be better of the one of DFEG, as predicted by the theory, its performance varies greatly in relation to $\eta$. On the other hand the performance of DFEG is close to the optimal one without the need to tune any parameters. It is also worth noting the catastrophic result we can get from a wrong tuning of $\eta$ in GD. Similar considerations hold for the *cpusmall* dataset in Figure 1(right).

## 6 Discussion

I have presented a new algorithm for online learning, the first one in the family of exponentiated gradient to be dimension-free. Thanks to new analysis tools, I have proved that DFEG attains a regret bound of $\mathcal{O}(U \log(U\,T + 1))\sqrt{T})$, without any parameter to tune. I also proved a lower bound that shows that the algorithm is optimal up to $\sqrt{\log T}$ term for linear and Lipschitz losses.

The problem of deriving a regret bound that depends on the sequence of the gradients, rather than on the $\boldsymbol{x}_t$, remains open. Resolving this issue would result in the tighter $\mathcal{O}(\sqrt{\sum_{t=1}^{T} \ell_t(\langle \boldsymbol{w}_t, \boldsymbol{x}_t \rangle)})$ regret bounds in the case that the $\ell_t$ are smooth [18]. The difficulty in proving these kind of bounds seem to lie in the fact that (2) is negative only because $H_t - H_{t-1}$ is bigger than $\|\boldsymbol{z}_t\|^2$.

### Acknowledgments

I am thankful to Jennifer Batt for her help and support during the writing of this paper, to Nicolò Cesa-Bianchi for the useful comments on an early version of this work, and to Tamir Hazan for his writing style suggestions. I also thank the anonymous reviewers for their precise comments, which helped me to improve the clarity of this manuscript.

## Footnotes

[1]The algorithm should be more correctly called Follow the Regularized Leader, however here I follow Shalev-Shwartz in [16], and I will denote it by OMD.

[2]All the theorems hold also in general Hilbert spaces, but for simplicity of exposition I consider a Euclidean setting.

[3]Despite what claimed in Section 1 of [19], the use of the time-varying regularizer $f_t(\boldsymbol{w}) = \frac{\sqrt{t}}{\eta}\|\boldsymbol{w}\|^2$ guarantees a sublinear regret for unconstrained online convex optimization, for any $\eta > 0$.

[4]http://www.csie.ntu.edu.tw/~cjlin/libsvmtools/datasets/

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
