[Supplementary Material]

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

$$\sum_{t=1}^{T} \Delta_t = f_T^*(\boldsymbol{\theta}_{T+1}) - f_0^*(\boldsymbol{\theta}_1) = f_T^*(\boldsymbol{\theta}_{T+1}) \geq \langle \boldsymbol{u}, \boldsymbol{\theta}_{T+1} \rangle - f_T(\boldsymbol{u}) = \sum_{t=1}^{T} \langle \boldsymbol{u}, \boldsymbol{z}_t \rangle - f_T(\boldsymbol{u}) .$$

Hence, for any sequence of vectors $\boldsymbol{w}_t'$ we have

$$\sum_{t=1}^{T} \langle \boldsymbol{u} - \boldsymbol{w}_t', \boldsymbol{z}_t \rangle \leq f_T(\boldsymbol{u}) + \sum_{t=1}^{T} \left( \Delta_t - \langle \boldsymbol{w}_t', \boldsymbol{z}_t \rangle \right) .$$

For the second statement, using the multivariate version of the Taylor's theorem, and recalling that $\boldsymbol{\theta}_{t+1} = \boldsymbol{\theta}_t + \boldsymbol{z}_t$ and $\boldsymbol{w}_t = \nabla f_t^*(\boldsymbol{\theta}_t)$, we have that

$$f_t^*(\boldsymbol{\theta}_{t+1}) - f_t^*(\boldsymbol{\theta}_t) \leq \langle \nabla f_t^*(\boldsymbol{\theta}_t), \boldsymbol{z}_t \rangle + \frac{\max_{0 \leq \tau \leq 1} \|\nabla^2 f_t^*(\boldsymbol{\theta}_t + \tau \boldsymbol{z}_t)\|}{2} \|\boldsymbol{z}_t\|^2$$

$$\leq \langle \boldsymbol{w}_t, \boldsymbol{z}_t \rangle + \frac{\lambda_t}{2} \|\boldsymbol{z}_t\|^2 .$$

Hence

$$\Delta_t = f_t^*(\boldsymbol{\theta}_{t+1}) - f_t^*(\boldsymbol{\theta}_t) + f_t^*(\boldsymbol{\theta}_t) - f_{t-1}^*(\boldsymbol{\theta}_t) \leq f_t^*(\boldsymbol{\theta}_t) - f_{t-1}^*(\boldsymbol{\theta}_t) + \langle \boldsymbol{w}_t, \boldsymbol{z}_t \rangle + \frac{\lambda_t}{2} \|\boldsymbol{z}_t\|^2 . \square$$

## 7.2 Proof of Lemma 2

*Proof.* The first two results comes directly from the definition of Fenchel dual.

To prove the second result, it is sufficient to prove that $\langle \nabla^2 f^*(\boldsymbol{\theta})\boldsymbol{x}, \boldsymbol{x} \rangle \leq \frac{1}{\beta \min(\|\boldsymbol{\theta}\|, \alpha)} \exp(\frac{\|\boldsymbol{\theta}\|}{\alpha}) \|\boldsymbol{x}\|^2$, $\forall \boldsymbol{x} \in \mathbb{X}$. For simplicity of exposition and w.l.o.g. assume the space $\mathbb{X} = \mathbb{R}^d$. Hence, we define $\Psi(x) = \frac{\alpha}{\beta} \exp(\frac{\sqrt{x}}{\alpha})$, and $\phi(x) = x^2$, so that

$$\langle \nabla^2 f^*(\boldsymbol{\theta})\boldsymbol{x}, \boldsymbol{x} \rangle = \Psi'' \left( \sum_{r=1}^{d} \phi(\theta_r) \right) \left( \sum_{r=1}^{d} \phi'(\theta_r) x_r \right)^2 + \Psi' \left( \sum_{r=1}^{d} \phi(\theta_r) \right) \sum_{r=1}^{d} \phi''(\theta_r) x_r^2 .$$

The first and second derivatives of $\Psi(x)$ and $\phi(x)$ are

$$\Psi'(x) = \frac{1}{2\beta\sqrt{x}} \exp \left( \frac{\sqrt{x}}{\alpha} \right), \qquad \Psi''(x) = \frac{\sqrt{x} - \alpha}{4\alpha\beta x^{\frac{3}{2}}} \exp \left( \frac{\sqrt{x}}{\alpha} \right),$$

$$\phi'(x) = 2x, \qquad\qquad\qquad \phi''(x) = 2,$$

hence we have

$$\langle \nabla^2 f^*(\boldsymbol{\theta})\boldsymbol{x}, \boldsymbol{x} \rangle = \left( 2 \sum_{r=1}^{d} \theta_r x_r \right)^2 \frac{\|\boldsymbol{\theta}\| - \alpha}{4\alpha\beta\|\boldsymbol{\theta}\|^3} \exp \left( \frac{\|\boldsymbol{\theta}\|}{\alpha} \right) + \frac{1}{\beta\|\boldsymbol{\theta}\|} \exp \left( \frac{\|\boldsymbol{\theta}\|}{\alpha} \right) \sum_{r=1}^{d} x_r^2 .$$

If $\|\boldsymbol{\theta}\| > \alpha$ we can use Cauchy-Schwarz inequality to have

$$\langle \nabla^2 f^*(\boldsymbol{\theta})\boldsymbol{x}, \boldsymbol{x} \rangle \leq \|\boldsymbol{x}\|^2 \frac{\|\boldsymbol{\theta}\| - \alpha}{\alpha\beta\|\boldsymbol{\theta}\|} \exp \left( \frac{\|\boldsymbol{\theta}\|}{\alpha} \right) + \frac{1}{\beta\|\boldsymbol{\theta}\|} \exp \left( \frac{\|\boldsymbol{\theta}\|}{\alpha} \right) \|\boldsymbol{x}\|^2 = \frac{1}{\alpha\beta} \exp \left( \frac{\|\boldsymbol{\theta}\|}{\alpha} \right) \|\boldsymbol{x}\|^2 .$$

On the other hand, if $\|\boldsymbol{\theta}\| \leq \alpha$, we get $\langle \nabla^2 f^*(\boldsymbol{\theta})\boldsymbol{x}, \boldsymbol{x} \rangle \leq \frac{1}{\beta\|\boldsymbol{\theta}\|} \exp(\frac{\|\boldsymbol{\theta}\|}{\alpha}) \|\boldsymbol{x}\|^2$. Putting together these two upper bounds, we have the stated result. $\square$

### 7.3 Proof of Lemma 3

The lemma upper bounds the exponential function with a quadratic form without the linear term.

*Proof.* Define $f(x) = x - \log\left(p + \frac{\exp(M)-p}{M^2}x^2\right)$, we will prove the equivalent statement $f(x) \leq 0$, $\forall 0 \leq x \leq M$. We have that $f(x)$ is equal to $0$ for $x = M$, hence it is enough to prove that the derivative of $f(x)$ is non-negative for $0 \leq x \leq M$. The derivative of $f(x)$ is

$$1 - \frac{2x\frac{\exp(M)-p}{M^2}}{p + \frac{\exp(M)-p}{M^2}x^2} = \frac{p + \frac{\exp(M)-p}{M^2}x^2 - 2x\frac{\exp(M)-p}{M^2}}{p + \frac{\exp(M)-p}{M^2}x^2},$$

that is always non-negative with the stated conditions on $p$. $\qquad\square$

### 7.4 Proof of Lemma 4

The lemma is a generalization of the well-known bound $\exp(x) \leq 1 + x + (e-2)x^2$, $\forall\, 0 \leq x \leq 1$.

*Proof.* Define $f(x) = x - \log\left(1 + x + \frac{\exp(M)-1-M}{M^2}x^2\right)$, we will prove the equivalent statement $f(x) \leq 0$, $\forall 0 \leq x \leq M$. We have that the $f(x)$ is equal to $0$ for $x = M$ and $x = 0$, hence it is enough to prove that $f(x)$ has only one minimum $0 < x < M$. The derivative of $f(x)$ is

$$1 - \frac{1 + 2x\frac{\exp(M)-1-M}{M^2}}{1 + x + \frac{\exp(M)-1-M}{M^2}x^2} = \frac{x\left(1 - 2\frac{\exp(M)-1-M}{M^2} + \frac{\exp(M)-1-M}{M^2}x\right)}{1 + x + \frac{\exp(M)-1-M}{M^2}x^2}.$$

Note that $2 - \frac{M^2}{\exp(M)-1-M} = 2\frac{\exp(M)-1-M-\frac{M^2}{2}}{\exp(M)-1-M} \geq 0$ because $\exp(M) \geq 1 + M + \frac{M^2}{2}$, by the Taylor expansion of exp. Hence, we have that the function $f(x)$ has a minimum in $x = 2 - \frac{M^2}{\exp(M)-1-M}$. $\qquad\square$

### 7.5 Proof of Lemma 5

*Proof.*

$$\frac{2}{\sqrt{p}} - \frac{2}{\sqrt{p+q}} = 2\frac{\sqrt{p+q}-\sqrt{p}}{\sqrt{p}\sqrt{p+q}} \geq \frac{q}{\sqrt{p}(p+q)} \geq \frac{q}{(p+q)^{\frac{3}{2}}}, \tag{6}$$

where in the first inequality we used the concavity of the square root function, and in the second we upper bounded $p$ with $p + q$. $\qquad\square$

### 7.6 Proof of Theorem 2

We first present a new tight lower bound on the Bernoulli distribution's tail, to allow to improve the leading constant of the regret lower bound and that has the right dependency in the denominator on $k$. A similar bound has been used in the proof of the lower bound in [23], but their proof does not explain how they numerically calculated the global minimum of function with an infinite number of local minima. We recover essentially their same bound for $k = \sqrt{T}$, and greatly improves it for bigger values. We also improve over similar known results, e.g. Proposition 7.3.2 in [13]. A similar bound for $k = \sqrt{T}$ for a Bernoulli with arbitrary probability appears also in [4], but they present a lower bound that is worse by a factor of about 2 compared to mine.

**Lemma 6.** *Let $T \geq 2$ an even number of Bernoulli random variables $z_i$. Then for any $k \in \mathbb{N}_0$ such that $k \leq \frac{1}{2}T - 1$, we have*

$$P\left(\sum_{i=1}^{T} z_i \geq \frac{1}{2}T + k\right) \geq \frac{\sqrt{2\pi}}{2\exp\left(\frac{1}{6}\right)} \frac{2^{-y^2}}{(\pi-1)y + \sqrt{y^2 + 2\pi}},$$

*where $y = \frac{2k}{\sqrt{T}}$.*

*Proof.* We use Theorem 2 in [14], that specialized to our case says that

$$P\left(\sum_{i=1}^{T} z_i \geq \frac{1}{2}T + k\right) \geq \frac{1}{2}\sqrt{T}\binom{T-1}{\frac{1}{2}T + k - 1} 2^{1-T} \frac{Q(y)}{\phi(y)}, \tag{7}$$

where $\phi(x)$ is the unit variance, zero mean Gaussian, $\frac{1}{\sqrt{2\pi}}\exp(-\frac{x^2}{2})$ and $Q(x)$ is its CDF, $\int_x^{+\infty} \phi(u)du$.

We start lower bounding the ratio $\frac{Q(y)}{\phi(y)}$ using the inequality in [3], that says

$$\exp\left(\frac{x^2}{2}\right) \int_x^{+\infty} \exp\left(-\frac{t^2}{2}\right) dt \geq \frac{\pi}{(\pi-1)x + \sqrt{x^2 + 2\pi}} .$$

To bound the binomial coefficient we make use of the following Stirling approximation, for any $n \geq 1$,

$$\sqrt{2\pi n}\, n^n \exp(-n) < n! < \exp\left(\frac{1}{12}\right)\sqrt{2\pi n}\, n^n \exp(-n) .$$

Hence, for any $n > 1$ and $p \in (0,1)$ such that $n - pn \geq 1$, after some algebra we obtain

$$\binom{n}{pn} \geq 2\frac{2^{n\,h(p)}}{\exp\left(\frac{1}{6}\right)\sqrt{2\pi n}},$$

where $h(p)$ is the binary entropy $-p\log_2 p - (1-p)\log_2(1-p)$. Using the well-known lower bound for the binary entropy

$$-p\log_2 p - (1-p)\log_2(1-p) \geq 1 - 4\left(p - \frac{1}{2}\right)^2,$$

and some overapproximation, we obtain

$$\binom{T-1}{\frac{1}{2}T + k - 1} = \binom{T}{\frac{1}{2}T + k}\frac{\frac{1}{2}T + k}{T} \geq \frac{1}{2}\binom{T}{\frac{1}{2}T + k} \geq \frac{2^{T - \frac{4k^2}{T}}}{\exp\left(\frac{1}{6}\right)\sqrt{2\pi T}}, \tag{8}$$

where we assumed that $\frac{1}{2}T - k \geq 1$. Putting together (7)-(8), and using the definition of $y$ we have

$$P\left(\sum_{i=1}^{T} z_i \geq \frac{1}{2}T + k\right) \geq \frac{\sqrt{2\pi}}{2\exp\left(\frac{1}{6}\right)} \frac{2^{-y^2}}{(\pi-1)y + \sqrt{y^2 + 2\pi}} . \qquad \square$$

*Proof of Theorem 2.* The first part of the theorem is proved observing that, if the algorithm guarantees a zero regret against a null competitor, this implies that $\sum_{t=1}^{T}\langle w_t, x_t\rangle \geq 0$ for any sequence of $x_t$. This implies that $w_t$ must be equal to the null vector, for all the $t$, hence the regret against competitors different from the null ones must be $\Omega(T)$.

For the second part of the theorem, we proceed similarly as in proof of Theorem 7 in [23], extending it to arbitrary vector spaces. Set $z_t = b_t q$, where $q$ is a fixed arbitrary vector in $\mathcal{X}$ with unitary norm, and $b_t$ are independent random variable that assumes the value of 1 with probability 0.5 and -1 with probability 0.5. Hence, we have that $\mathbb{E}[\sum_{t=1}^{T}\langle w_t, z_t\rangle] = 0$, and also $\sum_{t=1}^{T}\langle w_t, z_t\rangle \geq -\epsilon$ for the hypothesis on the regret. For any $k > 0$, it follows that

$$0 = \mathbb{E}\left[\sum_{t=1}^{T}\langle w_t, z_t\rangle\right] = \mathbb{E}\left[\sum_{t=1}^{T}\langle w_t, z_t\rangle \,\middle|\, \left\|\sum_{t=1}^{T} z_t\right\| < 2k\right] P\left(\left\|\sum_{t=1}^{T} z_t\right\| < 2k\right)$$

$$+ \mathbb{E}\left[\sum_{t=1}^{T}\langle w_t, z_t\rangle \,\middle|\, \left\|\sum_{t=1}^{T} z_t\right\| \geq 2k\right] P\left(\left\|\sum_{t=1}^{T} z_t\right\| \geq 2k\right)$$

$$\geq -\epsilon + \mathbb{E}\left[\sum_{t=1}^{T}\langle w_t, z_t\rangle \,\middle|\, \left\|\sum_{t=1}^{T} z_t\right\| \geq 2k\right] P\left(\left\|\sum_{t=1}^{T} z_t\right\| \geq 2k\right),$$

hence

$$\mathbb{E}\left[\sum_{t=1}^{T}\langle \boldsymbol{w}_t, \boldsymbol{z}_t\rangle \,\middle|\, \left\|\sum_{t=1}^{T}\boldsymbol{z}_t\right\| \geq 2k\right] \leq \frac{\epsilon}{P\left(\left\|\sum_{t=1}^{T}\boldsymbol{z}_t\right\| \geq 2k\right)} = \frac{\epsilon}{P\left(\left|\sum_{t=1}^{T}b_t\right| \geq 2k\right)}$$

$$= \frac{\epsilon}{2P\left(\sum_{t=1}^{T}b_t \geq 2k\right)}.$$

Using the fact that $P\left(\sum_{t=1}^{T}b_t \geq 2k\right) = P\left(\sum_{t=1}^{T}\frac{b_t+1}{2} \geq \frac{1}{2}T + k\right)$, where $\frac{b_t+1}{2}$ are Bernoulli random variables, we can apply Lemma 6, to obtain

$$P\left(\sum_{t=1}^{T}b_t \geq 2k\right) \geq \frac{\sqrt{2\pi}}{2\exp\left(\frac{1}{6}\right)}\frac{2^{-y^2}}{(\pi-1)y + \sqrt{y^2 + 2\pi}} \geq \frac{\sqrt{2\pi}}{2\exp\left(\frac{1}{6}\right)}\frac{2^{-y^2}}{\pi y + \sqrt{2\pi}} \geq \frac{1}{3}\frac{2^{-y^2}}{y+1},$$

where $y = \frac{2k}{\sqrt{T}}$. Set $k = \left\lfloor \frac{1}{2}\sqrt{\frac{T}{\log 2}\log\left(\frac{\eta\|\boldsymbol{u}\|\sqrt{T}}{3\epsilon}\right)} \right\rfloor$, where $0 < \eta < 1$. There exists a $T_0$ such that for $T \geq T_0$ we have that $y = \frac{2k}{\sqrt{T}} \geq 1$, hence $y + 1 \leq 2y$, so we have

$$\mathbb{E}\left[\sum_{t=1}^{T}\langle \boldsymbol{w}_t, \boldsymbol{z}_t\rangle \,\middle|\, \left\|\sum_{t=1}^{T}\boldsymbol{z}_t\right\| \geq 2k\right] \leq 3\,\epsilon\,2^{y^2}\,y \leq 2\eta k\|\boldsymbol{u}\|$$

From the above inequality we can infer that there exists a sequence of $\boldsymbol{z}_t$, and $\boldsymbol{u} = \alpha\sum_{t=1}^{T}\boldsymbol{z}_t$ with $\alpha > 0$, such that

$$Regret(\boldsymbol{u}) = \|\boldsymbol{u}\|\left\|\sum_{t=1}^{T}\boldsymbol{z}_t\right\| - \sum_{t=1}^{T}\langle \boldsymbol{w}_t\boldsymbol{z}_t\rangle \geq (1-\eta)\|\boldsymbol{u}\|2k\,. \qquad \square$$