[Reviews · NeurIPS 2013]

Submitted by Assigned_Reviewer_4

The paper considers unconstrained online convex optimization with general convex loss functions. It is shown that an appropriate version of the follow the regularized leader (FTRL) method achieves an almost optimal O(U \log (UT+1) \sqrt{T}) regret in the case when the norm U of the competitor is not known in advance (T is the time horizon). A lower bound shows that in the case of linear losses, the regret of any algorithm, whose regret is constant relative to the 0 predictor, is at least O(U \sqrt{T\log(U\sqrt{T})}).

The paper extends recent results of Streeter and McMahan [21], who considered the same problem, but their algorithms are more ad-hoc. The algorithm of the current paper is especially more appealing in high dimensions, since [21] runs a separate algorithm for each coordinate, and this paper only requires computation of inner products. In my view, the main advantage of this paper is the elegance of the provided solution, and how it follows nicely from tweaking the existing (standard) machinery. I also like the choice of the regularizers. An interesting feature of the algorithm, which the authors call "dimension-free", is that the (exponential) weights the algorithm keeps are set together (it would be good though if the authors could fit a short paragraph explaining how this results in a "free" kernelized version of the algorithm).

Some minor issues:
- Mirror descent (MD) vs FTRL: while Algorithm 2 is sometimes incorrectly called MD in the literature, it is in fact an FTRL algorithm for linearized losses.
- Section 2: the set X must have some unmentioned properties (e.g., existence of inner products). The inner products and vector products are used together (which is correct), but it might be better to use only one of them. It would be worth to add the expression for the gradient of f^* (it is useful in seeing that the algorithm is FTRL)
- Algorithm 1: step 7: \theta in the exponent should be \theta_t
- Theorem 1: define Lipschitz, y_t is not needed in the theorem (it is included in \ell_t)
- Please compare Theorem 1 to the Vovk-Azuri-Warmuth forecaster.
- Theorem 2: X must be a non-trivial vector space. It should also be clarified in the discussion around lines 172-178 that for the quadratic regularizer, FTRL achieves an O(\sqrt{T}) regret against the zero predictor, which cancels the \log\sqrt{T} term from the lower bound of Theorem 2 (hence not contradicting the upper bound in line 172).
- Lemma 1: the local norm is not defined. Maybe compare the lemma to Lemma 1 in http://homes.di.unimi.it/~cesabian/Pubblicazioni/genOmd.pdf (although it is not published yet).
- Discussion from line 233: maybe mention first how Alg. 1 is obtained from Alg. 2: otherwise the discussion is somewhat hard to follow.
- Lemma 2: define what norm is used for the Hessian
- line 268: w should be \theta
- line 285: Omit "Analogously" from the beginning of the sentence: the derivation of the second bound is absolutely different (in my opinion).
- line 415-416: Maybe the statement "the performance DFEG is close to the optimal one" is an exaggeration (if we compare, e.g., 4000 to 2000)
- First line of Lemma 5: Rephrase...
- line 624-625: I can hardly imagine that T/2+\sqrt{Tk}/2 would be an integer with the given choice of k (an assumption of Lemma 5). So please make the derivation precise.

Summary: The paper analyzes the problem of unconstrained online optimization. A "follow the regularized leader"-type algorithm is given (with time-varying regularizer) to tackle the problem, with upper bounds on its regret, together with almost matching lower bounds. The results follow nicely by tweaking the existing FTRL framework.

Submitted by Assigned_Reviewer_5

This paper provides a new algorithm and its analysis for the online learning problem. A learner receives a vector at each time step and has to select a label, after which the true label is received and the learner suffers some loss based on a convex and Lipschitz loss function that may vary over time. The algorithm is a variant of EG, with time-varying learning rates. The main advantage of the new regret bound is that the dimension of the vector space does not appear in the bound, and hence it holds for even infinite dimensional spaces. The novelty of the analysis is that, instead of discarding a negative term in the bound, it is used to balance out a term that might scale with the dimension.

The paper is well-written, the proofs seem sound. The result is of great interest in the machine learning community, and the new proof technique might be helpful for future work.

Question about the main theorem: the bound seems to be monotone increasing in "a". Why isn't the theorem (and the algorithm) stated with a=0.882 then?

Minor comments:
Throughout the paper: please consider not using \left and \right before parentheses in inline expressions. It would improve readability.
Line 7 of Alg. 1: In time step 1, you divide by zero.
Summary: Great new result in online learning with a version of EG with time-varying learning rates: the regret does not depend on the dimension of the space.

Submitted by Assigned_Reviewer_6

SUMMARY AND RELATED WORKS

This paper is about online linear regression with Lipschitz losses over L2-balls. In this setting, the authors introduce and study a new online algorithm that achieves a regret bound of the order of U*sqrt{T}*log(UT) after T rounds over any L2-ball of radius U, without knowing U in advance. This algorithm is close in spirit to that of Streeter and McMahan [21] (NIPS 2012), with however an interesting important difference: the regret bound of Theorem 1 is dimension-free in the sense that it does not depend on the dimension d of the input vectors x_t (note however that the algorithm of Streeter and McMahan was primarily designed for the L1-norm rather than for the L2-norm). This dimension-free bound suggests that the algorithm could be used in infinite dimension, which is useful for kernel regression. Another related algorithm is the standard gradient descent (see, e.g., Kivinen And Warmuth 1997 and Cesa-Bianchi 1999). Gradient descent can also achieve a dimension-free regret bound of the order of U*sqrt{T}, but at the price of knowing U beforehand. In the more realistic case when U is unknown, the regret bound of Gradient Descent has a worse dependency in U (U^2*sqrt{T} instead of U*sqrt{T}). On the contrary, the algorithm introduced in the present paper is adaptive to the unknown radius U (up to log factors). In a second part of the paper, the authors prove that their regret bound is optimal up to log factors, but this is in a rather weak sense (see below). They finally carry out some simulations to compare their algorithm to Gradient Descent with different values of the regularization parameter.


OVERALL ASSESSMENT

My overall opinion about the paper is positive for the following reasons:

PROS

1) To the best of my knwoledge, I think that Algorithm 1 is the first one that achieves (up to log factors) a regret bound of order U sqrt{T} without knowing U beforehand, via a sequential time-varying tuning of its parameters. Since the rate U sqrt{T} is optimal for any fixed U and for linear losses (at least for algorithms that don't have access to x_t, cf. [1]), and since the ideas underlying Algorithm 1 are far from trivial, Theorem 1 should be of interest to quite a few people in the machine learning community.
2) Comparisons to the literature are precise and honnest, which is a nice quality.

CONS

Mathematical concerns and significance:
3) Proof of Theorem 1: the proof relies on Lemma 1, where it is assumed that the functions f^*_t are twice differentiable. Since this is not the case for the functions used in Theorem 1 (because of the null vector), mathematical details are missing. In particular, what happens if the null vector lies in the segment [\theta_t,\theta_{t+1}] at some round t? Please provide careful details about how you carry out the Taylor expansion of f^*_t around \theta_t.
4) Theorem 2:
a) The lower bound is not as strong as one could expect, since it only holds for a fixed vector u (whose norm is roughly of order sqrt{T}). What happens for other vectors u? It would be better to have some information, e.g., for the case when \norm[u] -> 0 or when \norm[u] -> +\infty.
b) In the proof of Theorem 2, I think that to get E[\sum_t w_t^T z_t] = 0 on line 594, the authors implicitely used the fact that w_t is independent from z_t. This is not the case for all algorithms in the online protocol described at the beginning of the paper (the learner can not only use the past examples, but he has also access to x_t before making his decision). Therefore, the lower bound of Theorem 2 holds on a smaller class of algorithms: those that are only allowed to access to x_t through the norm \norm[x_t] (since w_t and z_t are then indeed independent). Please write this explicitely.

Writing style:
5) As such, the proofs are very technical and suffer from a lack of intuitions. Furthermore, the proof style could really be improved: some explanations are missing (e.g., simple arguments to carry out the calculations) and some details are not at the right place (e.g., the particular shape of f_t^* should be made precise right after Lemma 1, otherwise, we have to wait until Lemma 2 and line 303 to understand the relationship between Algorithms 1 and 2).

OTHER COMMENTS
6) Absract: is Algorithm 1 really an extension of the Exponentiated Gradient algorithm? I think that it is simply similar in shape. The verb 'extends' is thus midleading.
7) Introduction: the second paragraph is probably difficult to read for the unfamiliar reader. Please state the online decision protocol and the definition of online mirror descent (even in an informal way).
8) Line 44: a division by sqrt{T} is missing in the Euclidean regularizer (or, better: the regret bound is O(u^2/eta + eta*T), which is perhaps more standard).
9) Line 82: add the reference [8] to [3,17].
10) The term 'sample' for x_t is not appropriate: you could use 'input vector/data/covariates' instead.
11) Beginning of Section 2: the space Y is just the real line (so write it explicitely), and X is a Euclidean vector space (maybe Hilbert space?). Furthermore, the notation w_t^T x_t seems only appropriate when X is finite-dimensional.
12) Section 3: since all references about convex analysis are on finite-dimensional vector spaces, the extension to infinite-dimensional spaces should be clarified (but I agree that it is very natural).
13) Line 175: the optimality of the rate U*sqrt{T} was only proved for algorithms that don't use the knowledge of x_t (cf. [1]).
14) Line 179: what about the upper bound on a?
15) Lemma 1: please write that the norm of the Hessian is its spectral norm.
16) Line 221: \norm[z_t]^2_{f_t} -> \norm[z_t]^2
17) Lines 239-241: please explain briefly or point to the proof.
18) Lemma 2: it only holds for \theta \neq 0 (see also Comment 3 above).
19) Line 269: w_r -> \theta_r
20) Proof of Theorem 1: to help the reader, you could explicitely structure the proof as follows: Case 1 = "\norm[\theta_t] > ...", Case 2 = "\norm[\theta_t] \leq ...", and explain right from the beginning which case corresponds to which term in the minimum of the bound of Lemma 1.
21) Line 324: Lemma 3 is used with p=2 a^2 AND M=1/a.
22) Line 343: this is an equality.
23) Line 361: The sentence 'We now use ...' should be postponed right after Equation (3).
24) Line 366: several typos: there are two L factors instead of one in the second term of the right-hand side; \beta -> \beta_t; a factor of L is missing after the equal sign.
25) The lower bound of Theorem 2 is for the linear loss functions \ell_t(x) = - x. Please write this explicitely.
26) Several typos: 'vectorial space" -> 'vector space' (several times in the paper), 'supplemental material' -> 'supplementary material' (idem), 'derivative' -> 'derivatives' (line 270), 'the f(x)' -> 'f(x)' (line 511).

----------------------------------------
POST-REBUTTAL ADDENDUM

I'd like to thank the authors for their feedback. Since they addressed my two main questions very precisely and ruled out concerns 3) and 4), I clearly recommend acceptance. When polished (through the reviewers' suggestions), the paper will be a very nice one.

NB: I noted two other typos: a ||w|| is missing after \alpha_t on line 303, and u_t should be replaced with z_t on line 496.
Summary: This paper offers a nice new result (Theorem 1) about adaptive tuning in online linear regression on L2-balls. It should be of interest to quite many people working in online convex optimization. Since my main concerns (3 and 4) were properly addressed in the author rebuttal, I clearly recommend acceptance.
Author Feedback

Author rebuttal: We thank the Reviewers for their very detailed and useful comments.
We will carefully take into account all their suggestions in order to improve the paper, and the clarity of the proofs.



Reviewer_5 and Reviewer_6:
"Theta_t can be the null vector; the null vector can lie in the segment [theta_t,theta_{t+1}]"

We apologize for the lack of clarity in Lemma 1.
It turns out that Lemma 1 can be stated in a stronger and even clearer way. In particular, the second term in the min in line 222 holds for any w_t, while only the first expression in the min requires the additional hypothesis on the twice differentiability on the line [theta_t,theta_{t+1}].
Hence, from the proof of Theorem 1, the hypothesis on line 305 makes f^*_t twice differentiable on the entire line [theta_t,theta_{t+1}], i.e. the null vector cannot be on the segment [theta_t,theta_{t+1}]. On the other hand, on rounds where the hypothesis on line 341 is true, w_t can be set to any vector with norm less than exp(||theta_t||/alpha_t)/beta_t. So, on the very first round, and on any round when ||theta_t|| is equal to zero, w_t can be simply set to 0.
All these missing details will be added to the paper.



Reviewer_5:
"Theorem 1 monotone in a"

We agree that in Theorem 1, as currently stated, we could remove "a".
However, a tighter bound can be given that shows all the trade-offs in terms of the parameter "a" (the constant "34" would be replaced in this case, see line 361).
We will make this clearer in the final version.



Reviewer_6:
"Lower bounds is not strong as one would expect"

We thank the Reviewer for pointing out that it should be clearer that the lower bound matches the setting of Algorithm 1. Indeed, Algorithm 1 only needs to access to the norm of x_t, rather than x_t itself, before making its decision. Moreover, the norm of u is not fixed in the lower bound, only its direction is equal to sum z_t, whose norm is of the order of sqrt(T).
Both aspects will be better explained in the final version.